# Few-Shot Knowledge Graph Completion Model Based on Relation Learning

**Weijun Li** [1,2,*]**, Jianlai Gu** [2]**, Ang Li** [2]**, Yuxiao Gao** [2] **and Xinyong Zhang** [2]

[1] The Key Laboratory of Images and Graphics Intelligent Processing of State Ethnic Affairs Commission, North Minzu University, Yinchuan 750021, China
[2] School of Computer Science and Engineering, North Minzu University, Yinchuan 750021, China
*   Correspondence: lwj@nmu.edu.cn

**Abstract:** Considering the complexity of entity pair relations and the information contained in the target neighborhood in few-shot knowledge graphs (KG), existing few-shot KG completion methods generally suffer from insufficient relation representation learning capabilities and neglecting the contextual semantics of entities. To tackle the above problems, we propose a Few-shot Relation Learning-based Knowledge Graph Completion model (FRL-KGC). First, a gating mechanism is introduced during the aggregation of higher-order neighborhoods of entities in formation, enriching the central entity representation while reducing the adverse effects of noisy neighbors. Second, during the relation representation learning stage, a more accurate relation representation is learned by using the correlation between entity pairs in the reference set. Finally, an LSTM structure is incorporated into the Transformer learner to enhance its ability to learn the contextual semantics of entities and relations and predict new factual knowledge. We conducted comparative experiments on the publicly available NELL-One and Wiki-One datasets, comparing FRL-KGC with six few-shot knowledge graph completion models and five traditional knowledge graph completion models for five-shot link prediction. The results showed that FRL-KGC outperformed all comparison models in terms of MRR, Hits@10, Hits@5, and Hits@1 metrics.

**Keywords:** knowledge graph; complete the knowledge graph; few-shot relation; neighborhood aggregation; link prediction

## 1. Introduction

The Knowledge Graph (KG) is a concept introduced by Google in 2012 to improve the speed of search engines 1. It contains rich and diverse relational data and is widely used in various production tasks in society. Existing knowledge graphs include Freebase 2, YAGO 3, NELL 4, and Wikidata 5. They all contain many triples formed by facts, which are usually represented in the form of (head entity, relation, tail entity), i.e., $(h, r, t)$.

In the real world, this graph-structured knowledge plays an important role in many downstream applications, such as semantic search 6, intelligent question answering 7, and personalized recommendations 8. However, knowledge graphs still suffer from the issue of incomplete facts. To address this problem, it is necessary to use Knowledge Graph Completion (KGC) to automatically infer and fill in missing facts, further enhancing the value of knowledge graphs.

In recent years, researchers have proposed many knowledge graph completion models based on knowledge graph embedding techniques 9 for the KGC task, including TransE 10, TransH 11, DistMult 12, ComplEx 13, and ConvE 14. These models have partially addressed the incomplete entity and relation problem in knowledge graphs. However, these embedding models usually focus on a small proportion of frequent relations, while in real-world knowledge graphs, most relations exhibit long-tailed distributions 15.

For example, in Wikidata, approximately 10% of relations have fewer than 10 triples. Furthermore, in most practical applications, such as recommendation systems and social media networks, knowledge graphs undergo dynamic changes over time. That is, a relation typically contains thousands of associated triples, while most relations only contain a few triples. In the practical context with only a few entities and relations, the performance of traditional knowledge graph completion models significantly declines.

To address the knowledge graph completion problem for uncommon entities and relations, researchers have proposed a variety of Few-Shot Knowledge Graph Completion (FKGC) methods. The GMatching model 16 was the first to be proposed for solving FKGC. It enhances entity embeddings using one-hop neighbor structures to improve the referenced semantic representation and employs a Long Short-Term Memory network (LSTM) to match the embedding representations with the target, obtaining similarity scores for relation prediction. Building on this, the FSRL model 17 and the FAAN model 18 use attention mechanisms to improve neighbor encoders. The MetaR model 19 solves this problem by passing meta information specific to the relations. The GANA model 20 addresses this issue by designing a gated and attention-based neighborhood aggregator and constructing a global-to-local framework to handle complex relations simultaneously.

In the FKGC task, the aforementioned models have achieved good results. However, these methods still have some limitations: (1) FAAN fails to effectively utilize the valuable information from high-order neighboring entities (the most relevant high-order neighborhood set in nonlocal graphs 21) and cannot discern the importance of information for different neighbors, leading to noise-related issues. (2) FSRL simply uses a recurrent autoencoder to aggregate a small reference set. However, during the training process, the FSRL model tends to depend excessively on entity embeddings, leading to overfitting of relations and a decline in the generalization capability of the model. (3) FSRL does not consider the translational property of the TransE model during matching queries, which can lead to a decline in matching accuracy. (4) Previous models have not adequately considered the impact of entity pairs on contextual semantics 21, resulting in reduced accuracy in link prediction. To overcome the limitations of existing methods, we propose utilizing high-order neighborhood entity information to represent each few-shot relation. By considering relations, our FKGC model can infer missing facts more effectively. This approach enhances the model's generalization capability and allows for the utilization of more contextual semantics to handle few-shot relations.

To improve the accuracy of link prediction, this paper proposes a few-shot knowledge graph completion model (FRL-KGC), which makes the following contributions:

(1) We introduce the FRL-KGC model, which incorporates a gating mechanism to extract valuable contextual semantics from the head entity, tail entity, and neighborhood information, specifically addressing high-order neighborhood information in the knowledge graph. Furthermore, we leverage the correlation between entity pairs in the reference set to represent relations, reducing the dependency of relation embeddings on the central entity.

(2) We effectively utilize both the structural and textual information of the knowledge graph to capture features related to few-shot relations.

(3) Experimental evaluations are conducted on two publicly available datasets, and the results demonstrate that our proposed model outperforms other KGC models. Additionally, ablation experiments validate the effectiveness of each key module in our model.

## 2. Related Work

Knowledge graph completion is the task of filling in missing entities, relations, and attributes in a knowledge graph through automatic inference and learning. In recent years, researchers have attempted to solve this problem from various perspectives. In traditional knowledge graph completion research, the mainstream approach is based on representation learning of knowledge graphs, which embeds the knowledge graph containing entities and relations into a low dimensional vector space, and represents their semantic features through spatial relations between vectors to discover potential connections 9. Currently, knowledge graph completion methods based on knowledge graph embeddings can be divided into translation-based methods, semantic matching-based methods, and neural network-based methods.

### 2.1. Translation-Based Methods

The approach based on translation methods treats relations as translation operations between entity pairs, where modeling relations is viewed as a form of translation in a low-dimensional entity representation space. The existence of associations between entities and relations is determined using a distance scoring function. TransE 10 is a popular translation model that regards a relation as a translation from the head entity to the tail entity. It can handle large-scale knowledge graph completion tasks but struggles with complex relation types. Therefore, researchers have proposed several improved translation models to address the "1–N", "N–1" and "N–N" problems, such as TransH 11, TransR 22, TransD 23, and TransG 24. Among them, TransH embeds each entity into different, relation-specific hyperplanes to address the complex relation representation problem; TransR first projects entities into the corresponding relation spaces and then establishes translation relations from head entities to tail entities. These improved methods have enhanced the performance and scalability of translation models.

### 2.2. Semantic Matching-Based Methods

Semantic matching-based methods use similarity-based scoring functions to mine the potential semantics between entities and relations, aiming to solve the knowledge graph completion task. Among them, the RESCAL model 25 represents entities as vectors and relations as matrices, using a bilinear function as the scoring function and obtaining prediction scores through tensor decomposition. However, as the embedding dimension increases, RESCAL faces the issues of parameter explosion and increased model complexity. To address these problems, the DistMult model 12 constrains the relation matrix to be diagonal, thereby simplifying the model, but this also results in an inability to handle asymmetric relations. The ComplEx model 13 uses complex vectors to represent entities and relations, which can effectively model various binary relations such as symmetric and asymmetric relations. Compared to translation-based methods, semantic matching-based methods have higher complexity, lower model-training efficiency, and weaker generalization ability, but they can better capture the implicit semantics between entities.

### 2.3. Neural Network-Based Methods

Neural network-based methods rely on the powerful learning and expressive capabilities of neural networks for modeling. ConvE, first introduced in 14, applies convolutional neural networks to knowledge graph completion, using two-dimensional convolution to vectorize entities and relations to concatenate them, ultimately obtaining the final embeddings through a fully connected layer and pooling. Compared to DistMult 12, ConvE achieves higher performance with fewer parameters. However, semantic information in knowledge graphs propagates along paths, and ConvE does not consider the importance of path information. RSN 26 takes into account path information in knowledge graphs, combining recurrent neural networks and residual learning, and captures relations between entities through random walks to improve inference effectiveness. With the

continuous expansion of the deep learning field and the popularity of graph neural networks, GATs 27 capture information by aggregating graph structured data in knowledge graphs, using the attention mechanism of graph attention networks to assign different weights to different neighboring nodes, thus capturing the most crucial neighboring node information.

*2.4. Few-Shot Learning*

Traditional knowledge graph completion methods require a large number of training instances to improve model accuracy. However, in the real world, there are numerous new facts, and knowledge graphs are constantly changing dynamically. When knowledge graphs cannot provide sufficient training instances for these new facts, the model's training process is greatly limited. Therefore, researchers have begun to explore knowledge graph completion tasks with only a small number of instances. In metric-learning-based methods, GMatching 16 was the first to propose the few-shot knowledge graph completion problem, obtaining embedding representations from one-hop neighbor structures in the neighborhood and using LSTM networks to match embeddings with targets, yielding similarity scores to measure the similarity between query triples and the reference set. However, GMatching does not distinguish between neighboring information in the neighborhood. FSRL 17 can effectively capture knowledge from heterogeneous graph structures, aggregate representations of a small number of samples, and assign different weights to neighborhood information using a heterogeneous neighbor decoder. Compared to the first two models, FAAN 18 considers the dynamic attributes of entities and relations and captures dynamic features that change in different tasks through attention mechanisms, thereby improving their fine-grained semantic representations. MetaR 19, in contrast to the above methods, which utilize neighborhood information to enhance entity embeddings, adopts a meta learning framework including gradient meta and relation meta for few-shot knowledge graph completion tasks, effectively improving the model's learning ability. GANA 20 improves upon MetaR, using a global local framework to accurately filter out noise information in the neighborhood and addressing the complex relation problems of one-to-many (1–N), many-to-one (N–1), and many-to-many (N–N) in knowledge graphs.

**3. Preliminaries**

In response to the mentioned problems, this paper proposes a Few-shot Relation Learning-based Knowledge Graph Completion model (FRL-KGC). For clarity, Table 1 presents common symbols used in this paper and their corresponding meanings.

**Table 1.** Symbol explanation table.

| Symbol | Description |
|---|---|
| $\mathcal{G}$ | knowledge graph |
| $\mathcal{G}'$ | background knowledge graph ($\mathcal{G}$ removes all subgraphs of task relations) |
| $\mathcal{E}, \mathcal{R}$ | entity and relation of a knowledge graph |
| $\mathcal{S}_r$ | reference set corresponding to relation $r$ |
| $\mathcal{Q}_r$ | query set corresponding to relation $r$ |
| $\mathcal{Q}_r^-$ | negative query set corresponding to relation $r$ |
| $\mathcal{R}_{br}$ | the relation set of background knowledge graph $\mathcal{G}'$ |
| $\mathcal{R}_{task}$ | task relation set |
| $C_{h_j,r}$ | candidate set for the potential tail entity of $\left(h_j, r, ?\right)$ |
| $T_{mtr}$ | set of meta train |
| $T_{mte}$ | set of meta test |

| | |
|---|---|
| $h,r,t$ | embedding features of fact triplets $h, r, t$ |
| $\mathcal{N}_h^o$ | the higher-order neighbor set of entity $h$ |
| $e$ | entity representation updated by neighborhood entity encoder |
| $r'$ | task relation representation updated by relation encoder |
| $z_i^k$ | representation of triples corresponding to relation $r$ |
| $\mathcal{L}$ | loss function |

### 3.1. Problem Formulation

A knowledge graph $\mathcal{G}$ is composed of various facts, each of which can be represented as a set of triples in the form $\mathcal{G} = \{(h,r,t) \subseteq \varepsilon \times \mathcal{R} \times \varepsilon\}$, where $\varepsilon$ and $\mathcal{R}$ represent the entity set and relation set, respectively. The problem of knowledge graph completion is to infer one fact by giving two facts in a triple. The focus of this study is predicting the tail entity $t$ by giving the head entity $h$ and relation $r$, which involves determining whether $(h,r,?)$ holds true. Different from general knowledge graph completion problems, few-shot knowledge graph completion problems are performed with a limited number of reference samples.

In summary, we provide the definition of a few-shot KGC problem as follows.

**Definition 1.** *Given a problem relation $r$ and its reference set $\mathcal{S}_r = \{(h_i,t_i) | (h_i,r,t_i) \in \varepsilon \times \mathcal{R} \times \varepsilon\}_i$, the tail entity $t_j$ is predicted based on the information provided by the knowledge graph $\mathcal{G}$ and $\mathcal{S}_r$, as well as the connection between the head entity $h_j$ and the problem relation $r$ in the query triple $(h_j,r,?)$. In this case, $|\mathcal{S}_r| = K$, where $K$ is typically a small value, hence the term few-shot knowledge graph completion.*

### 3.2. Few-Shot Learning Settings

According to Definition 1, few-shot knowledge graph completion is a relation-specific task. In the knowledge graph $\mathcal{G} = \{(h,r,t) \subseteq \varepsilon \times \mathcal{R} \times \varepsilon\}$, the relation $\mathcal{R}$ is divided into $\mathcal{R}_{br}$ and $\mathcal{R}_{task}$. $\mathcal{R}_{br}$ represents the relation set in the background knowledge graph $\mathcal{G}'$, which is a subgraph of the knowledge graph $\mathcal{G}$ obtained by removing all task relations. $\mathcal{R}_{task}$ represents the task relation set including $\mathcal{R}_{train}$, $\mathcal{R}_{validation}$, and $\mathcal{R}_{test}$, which are used in the meta training, meta validation, and meta testing stages of the FKGC task, respectively.

In the meta training stage, for each training task $r \in \mathcal{R}_{train}$, the associated triplets are randomly divided into a reference set $\mathcal{S}_r$, a query set $\mathcal{Q}_r$, and a set $\mathcal{D}_r = \{\mathcal{S}_r, \mathcal{Q}_r\}$. The reference set $\mathcal{S}_r = \{(h_i,t_i) | (h_i,r,t_i) \in \mathcal{G}\}_i$ contains $K$ entity pairs $(h_i,t_i)$. The query set $\mathcal{Q}_r = \left\{ \left( h_j, t_{true} / C_{h_j,r} \right) \right\}_j$ consists of the true tail entity $t_{true}$ of the query triplet and a candidate tail entity set $C_{h_j,r}$. The candidate tail entities for each triplet in the query set are constructed based on entity type constraints 18. This construction method ensures that the query triplets will be matched with semantically similar candidate tail entities. Finally, all the query tasks are combined into a set $T_{mtr} = \{\mathcal{D}_r\}$. For each query triplet $(h,r,?) \in \mathcal{Q}_r$, the similarity scores between the candidate entity pairs $(h_j, C_{h_j,r})$ and all reference entity pairs $(h_i,t_i)_i \in \mathcal{S}_r$ are computed. The candidate entity with the highest-ranking score is selected as the training result.

In the meta testing stage, the overall procedure is similar to the meta training stage. Firstly, the associated triplets of the test task $r^{'} \in \mathcal{R}_{test} \left( \mathcal{R}_{train} \cap \mathcal{R}_{test} = \varnothing \right)$ are randomly divided into a reference set $\mathcal{S}_r^{'}$ and a query set $\mathcal{Q}_r^{'}$ to define $\mathcal{D}_r^{'} = \left\{ \mathcal{S}_r^{'}, \mathcal{Q}_r^{'} \right\}$. Secondly, all the test query tasks are combined into a meta test set, denoted as $T_{mte} = \left\{ \mathcal{D}_r^{'} \right\}$. Finally, the candidate entities are scored, and the candidate entity with the highest score is selected as the predicted result of the model. In summary, the FKGC task involves ranking the true tail entity $t_j$ and candidate tail entity $t^{'} \in C_{h_j, r}$ of a triple $\left( h_j, r, ? \right)$. Given the task relations in set $r$, the information provided by the knowledge graph $\mathcal{G}$, and the reference set $\mathcal{S}_r$, the objective is to ensure that the rank of $t_j$ is higher than that of all the candidate entities.

Figure 1 illustrates an example of a five-shot KGC task. In the few-shot KGC task, the reference set for the query triplet (Windows, ProducedBy, ?) consists of five associated triplets. The goal is to use the reference set to match the correct tail entity for the query triplet (Windows, ProducedBy, ?). In this case, the true tail entity "Microsoft" should be ranked higher than other entities. The core of the FKGC task lies in predicting new facts with minimal reference information.

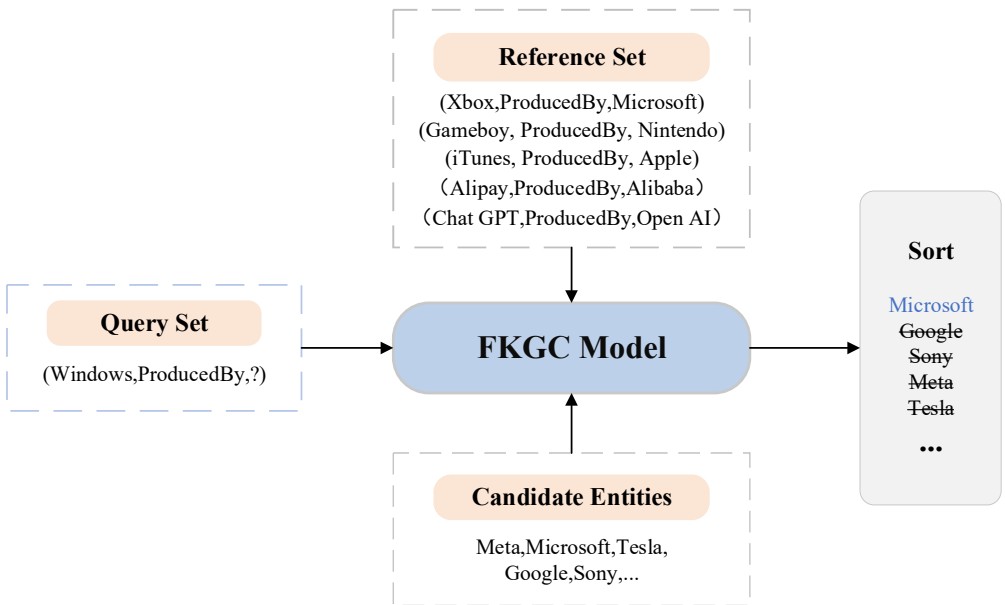

**Figure 1.** An example of a five-shot KGC task.

## 4. Model

FRL-KGC is a model that utilizes the background knowledge graph $\mathcal{G}^{'}$ and the structural information of the knowledge graph to train and learn for the task of few-shot tail entity prediction for relations. The overall framework of FRL-KGC is illustrated in Figure 2, and includes:

(a) High-order neighborhood entity encoder based on gate mechanism, which adaptively aggregates neighborhood information for entities.
(b) Relation representation encoder, which utilizes the relation information of reference entity pairs' neighbors to reduce the dependency of relations on entity embeddings and improve generalization.
(c) Transformer learner, which combines LSTM units and Transformer modules to further learn the representation of task relations.

(d) Matching process computation, which utilizes the semantic embeddings of relations outputted by the Transformer learner to calculate the similarity with the query relation, predicting new triplets.

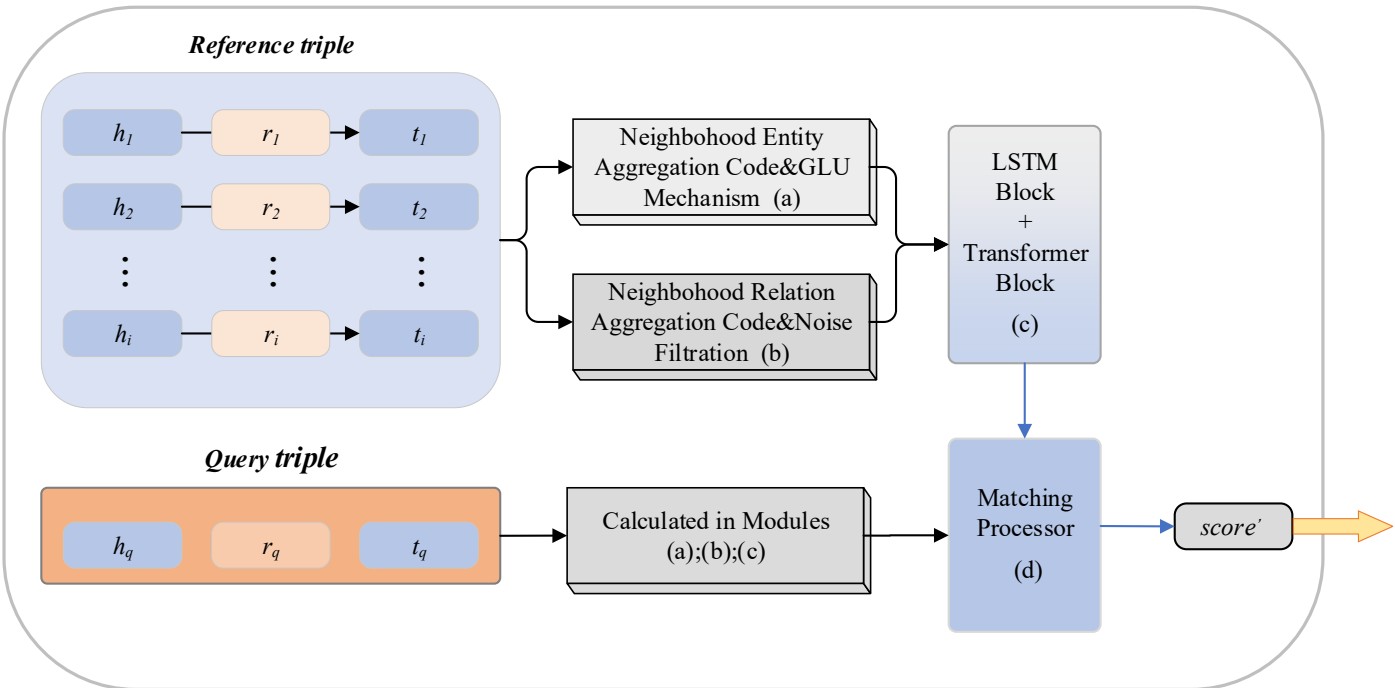

**Figure 2.** Overview of the FRL-KGC framework.

### 4.1. Entity Encoder Based on Gate Mechanism

In knowledge graphs, heterogeneous neighbors of entities have different impacts on the representation of the entities. Adaptively aggregating neighboring information according to the task relation $r$ can improve the quality of the central entity's representation 18. However, this method does not aggregate higher-order neighborhood entity information into the central entity and overlooks the effect of the ratio between useful neighbors and useless neighbors on the encoding of the central entity. Based on this, we designed a gated higher-order neighborhood entity encoder that extends the "adaptive neighborhood encoder" of the FAAN model and expands it to higher-order neighborhoods with the addition of a gating mechanism. This approach enhances the expressiveness of entities while reducing the impact of noisy neighbors on the updating of central entity encoding. The main structure of the gated higher-order neighborhood entity encoder is shown in Figure 3.

Given a task triplet $(h, r, t)$, assuming entity $h$ is the target entity, its encoding is updated through the higher-order neighborhood entity encoder. The higher-order neighborhood of entity $h$ is defined as $\mathcal{N}_h^o = \{(r_i^o, t_i^o) | (h, r_i^o, t_i^o) \in \mathcal{G}'\}$, where $\mathcal{G}'$ is the background knowledge graph, and $r_i^o$ and $t_i^o$ represent the *i-th* higher-order neighbor relation and the corresponding tail entity of entity $h$, respectively. To quantify the features of entity $h$, we first use the metric function $\varphi^o(\cdot)$ to calculate the similarity between the reference relation $r$ and the adjacent higher order relation $r_i^o$ of entity $h$, as shown in Equation (1).

$$\varphi^o\left(r, r_i^o\right) = r^T\left(W r_i^o + b\right) \tag{1}$$

where $r$ represents the initial feature of task relation $r$; $r_i^o$ represents the pre-trained embedding of higher-order adjacent relation $r_i^o \in \mathcal{N}_h^o$; $W \in \mathbb{R}^{d \times d}$ and $b \in \mathbb{R}^{d \times 1}$ are the weight matrix and bias parameters, respectively; and $\varphi^o(\cdot)$ is the bilinear dot product function.

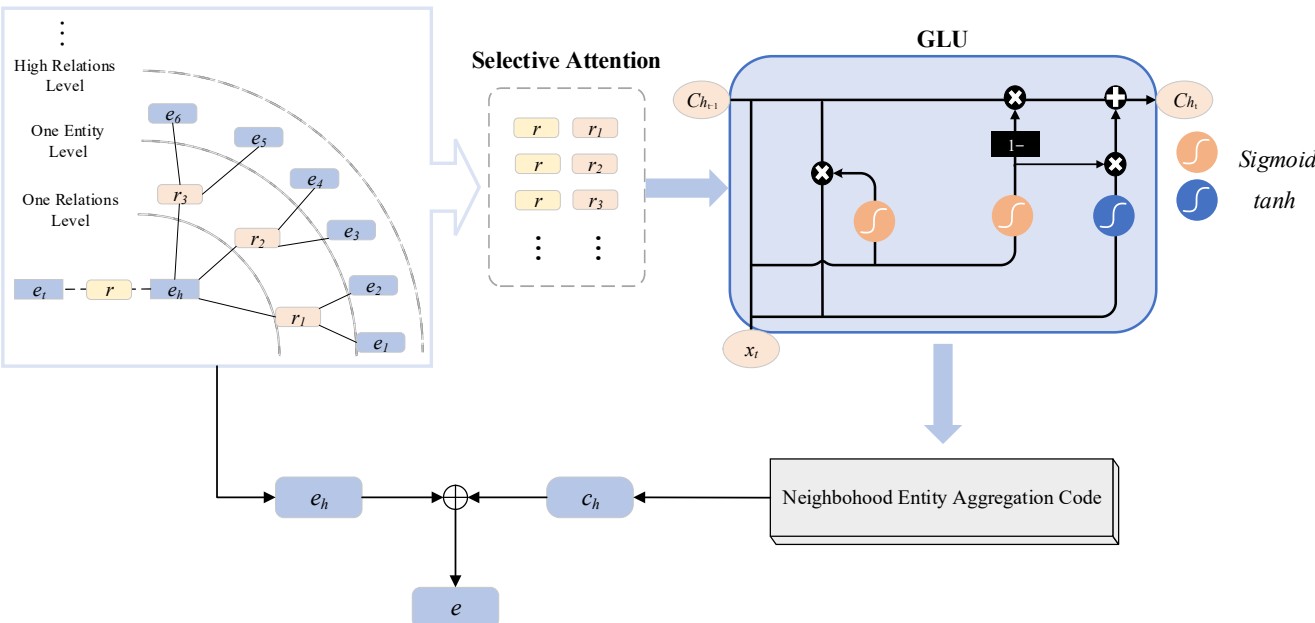

**Figure 3.** The main structure of the high-order neighborhood entity encoder based on the gating mechanism.

Next, based on the similarity score $\varphi^o(\cdot)$ of the higher-order neighborhood encoding, the attention mechanism is employed to assign higher attention scores to those entities $t_i^o$ with higher scores. To further improve the representation quality of entity $h$ and reduce the impact of noisy neighbors, higher weights are allocated to more important neighbors. The "Gating Mechanism" is introduced in the process of calculating neighbor weights to adaptively compute weight $\alpha_i$, as shown in Equations (2) and (3):

$$G\left(h^o, e_i^o\right) = \sum_{o=1}^{n} \left(h^o \cdot e_i^o\right) \cdot r_i^o \tag{2}$$

$$\alpha_i = softmax\left(G\left(h^o, e_i^o\right)\right) \tag{3}$$

The gating mechanism function $G(\cdot)$ calculates the inner product of the central entity and neighboring entities, performs matrix multiplication using the task relation $r_i^o$, and sums the results. Here, $n$ is the maximum neighboring order between the central entity $h^o$ and neighboring entity $e_i^o$, and $o$ represents the order number. It takes the central entity $h^o$ and related neighboring entities $e_i^o$ as inputs, and outputs a metric representing the relevance between the neighborhood and the central entity. $softmax(\cdot)$ converts the input into a probability distribution of neighbors.

To better capture the specific task relation, a learnable task relation matrix $R$ is introduced, which is used to update task relation $r_i^o$. The calculation process is shown in Equation (4):

$$r_i^{'o} = R \cdot r_i^o \tag{4}$$

where $r_i^{'o}$ is the updated task relation.

With this, the neighborhood encoding $c_h$ of the central entity $h$ can be obtained, as shown in Equation (5):

$$c_h = GLU\left( \sum_{o=1}^{n} softmax\left( G\left( h^o, e_i^o \right) \right) \cdot e_i^o \cdot r_i^{'o} \right) \tag{5}$$

where the function $GLU(\cdot)$ is the gated linear unit activation function.

Finally, the output of the neighborhood encoder and the initial feature of the central entity are used to adaptively update the entity representation, as shown in Equation (6):

$$e = \sigma\left( w_1 \cdot c_h + w_2 \cdot e_h \right) \tag{6}$$

where $\sigma(\cdot)$ represents the *sigmoid* activation function, $h$ represents the initial feature of the central entity, $w_1, w_2 \in \mathbb{R}^{d \times d}$ represent learnable parameters, and $e \in \mathbb{R}^{d \times 1}$ represents the updated entity embedding of the central entity $h$.

This entity update step is applied to all entities in the reference set and query set. The proposed higher-order neighborhood entity encoder in this paper takes into account the information implied by the neighbors of higher-order neighborhood entities and introduces a gating mechanism to filter noisy neighbors for specific task relations. The range of high-order neighborhoods is set to 3 in this method, as a range that is too low would lead to a loss of a significant amount of hidden information, while a range that is too high would cause a decrease in the model's performance.

*4.2. Relation Representation Encoder*

Through the above steps, the entities of the reference set and the query set are encoded to obtain their representations $h$ and $t$. Following the method in the FAAN paper 18, we can express the prototype relation $r_s$ of the reference set by connecting $h_s$ and $t_s$. As shown in Equation (7):

$$r_s \approx h_s + t_s \tag{7}$$

MetaR [19] only represents few-shot relations by averaging the embeddings of all entity pairs in the support set, without considering the correlation between entity pairs in the reference set. Additionally, in previous methods, FSRL [17] simply used a cyclic auto-encoder to aggregate a small reference set. However, as training deepens, the model's relation embeddings become overly reliant on entities, leading to relation overfitting and reduced generalization capability.

In this paper, we utilize the neighbor relations of entity pairs in the reference set to enrich the semantic representation of the current relation and reduce the reliance on relation embeddings for entities, thereby enhancing the model's generalization ability. The main structure of the relation representation encoder is shown in Figure 4.

In order to represent the neighbor relations of the head entity $h_i$ and tail entity $e_i$ of the *i-th* entity pair $\left( h_i, e_i \right)$ in the reference set, we define the reference set entity pair neighbor relation set $R_r^{'} = \mathcal{S}_r \left\{ r | \left( h_i, r, ? \right), \left( e_i, r, ? \right) \in \mathcal{G} \right\}$. To enhance clarity, we represent the relation $r$ in the set of neighbor relations as a vector, as shown in Equation (8):

$$E_{r_{s,i}} = \left\{ r, r \in \mathcal{S}_r \right\} \tag{8}$$

where $E_{r_{s,i}}$ represents the vectorized representation of the *i-th* neighbor relation $r_s$ in the set $R_r^{'}$.

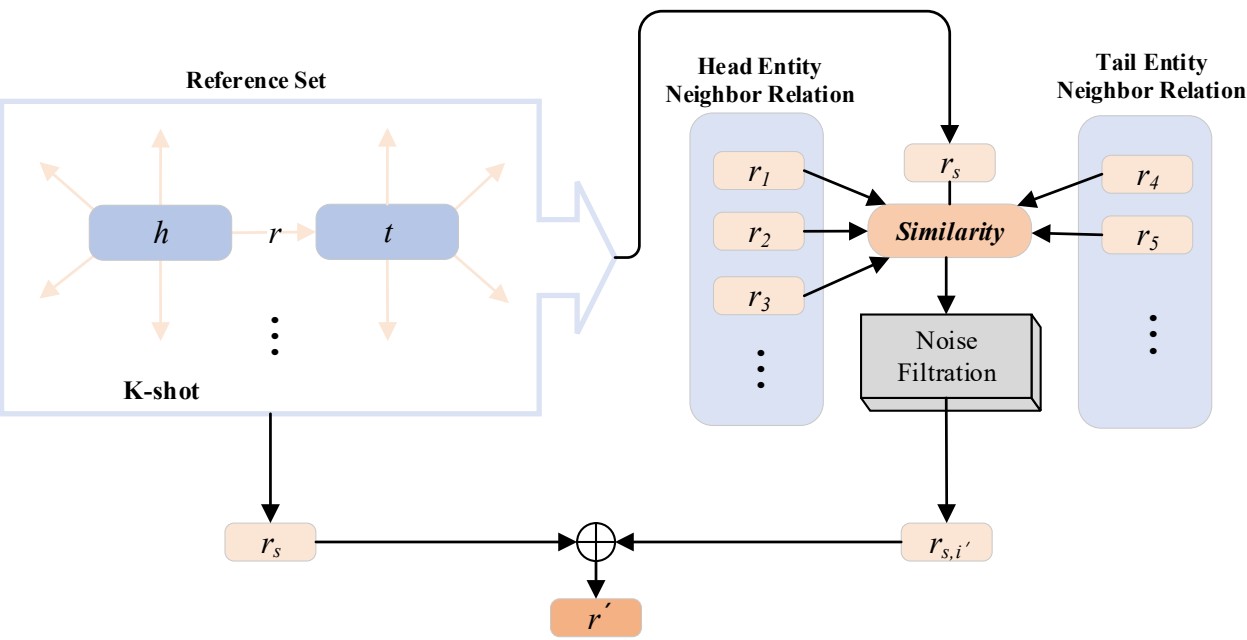

**Figure 4.** The main structure of the relation representation encoder.

To enrich the semantic expression of the current relation while reducing the impact of noise, the dot product similarity of feature vectors is used to calculate the similarity between $E_{r_{s,i}}$ and the reference set prototype relation $r_s$. The calculation process is shown in Equation (9):

$$E_{sim}(i) = r_s \cdot E_{r_{s,i}}^{T} \tag{9}$$

where $E_{sim}(i)$ represents the vector similarity score of the *i-th* neighbor relation $r_s$.

In this paper, we only retain the T neighbor relations with the highest similarity scores. Therefore, a noise filtering method is used to filter out vectors with lower similarity scores, as shown in Equation (10):

$$\boldsymbol{E}_{sim}^{'}(i) = \begin{cases} 1, & \text{if } E_{sim}(i) \in Top\left(\boldsymbol{E}_{sim}(i), T\right) \\ 0, & \text{Otherwise} \end{cases} \tag{10}$$

where the function $Top(\cdot)$ returns the *T* neighbor relation vectors with the highest similarity scores in $\boldsymbol{E}_{sim}(i)$ (T is set to 3). $\boldsymbol{E}_{sim}^{'}(i)$ represents the neighbor relation vector after noise filtering.

Next, the noise filtered neighbor relation vector $\boldsymbol{E}_{sim}^{'}(i)$ is encoded, as shown in Equations (11) and (12):

$$r_{s,i}^{'} = \sum W_{r_{s,i}} \cdot \boldsymbol{E}_{sim}^{'}(i) \tag{11}$$

$$W_{r_{s,i}} = \frac{\exp(\lambda \cdot \sigma(\boldsymbol{E}_{sim}^{'}(i)))}{\sum_{r_{s,i} \in R_r^{'}} \exp(\lambda \cdot \sigma(\boldsymbol{E}_{sim}^{'}(i^T)))} \tag{12}$$

where $r_{s,i}^{'}$ represents the representation after integrating the neighbor relation vector with the least noise in the reference set entity pairs; $W_{r_{s,i}}$ represents the weight during the encoding process; $\boldsymbol{E}_{sim}^{'}(i^{T})$ is the vector representation of each neighbor relation in the reference set entity pairs; $\lambda$ is a learning parameter; and $\sigma(\cdot)$ represents the activation function.

Finally, the prototype relations $r_{s}$ and $r_{s,i}^{'}$ of the reference set are merged, as shown in Equation (13):

$$r^{'} = \sigma\left(w_{1} \cdot r_{s} + w_{2} \cdot r_{s,i}^{'}\right) \tag{13}$$

where $\sigma(\cdot)$ represents the activation function, and $w_{1}, w_{2}$ denote the learnable parameters.

### 4.3. Transformer Learning Framework

In the FKGC task, the core objective is to incorporate as much semantic information as possible into the final output relation representation *R*, thereby enhancing the fine-grained semantics of different entity pairs in the few-shot reference set. Due to the powerful learning capability of Transformers [28], this paper utilizes a Transformer as a learner to further learn the relation representation of triplets. In order to obtain more accurate relation representations, the FRL-KGC model takes the entity embeddings from the high-order neighborhood entity encoder and the task relation embeddings from the relation representation encoder as inputs to the Transformer learner, enabling further learning of relation representations. Inspired by R-TLM (Recurrence Transformer Language Model) [29], this paper optimizes the learner based on a simplified R-TLM module. The main structure of the Transformer learner is illustrated in Figure 5.

The task relation $r$ and its corresponding entity pair $(h, t)$ are represented as a sequence $X = \{x_{1}, x_{2}, x_{3}\}$, where $x_{1}$ and $x_{3}$ represent the head and tail entities, and $x_{2}$ represents the task relation. Firstly, the input of the Transformer is defined as $z_{i}^{k}$, and for an element $x_{i}$ in the sequence $X$, it is represented as Equation (14):

$$z_{i}^{k} = x_{i}^{emb} + x_{i}^{pos} \tag{14}$$

where $x_{i}^{emb}$ represents the embedding of the element and $x_{i}^{pos}$ represents the positional embedding. The entity embeddings $x_{1}^{emb}$ and $x_{3}^{emb}$ are the updated entity representations from the high-order neighborhood entity encoder (Equation (6)), and the relation embedding $x_{2}^{emb}$ is obtained from the relation representation encoder (Equation (13)). Firstly, $z_{i}^{0}$ is inputted into the LSTM Block, and the hidden state is $z_{i}^{1}$. Next, the $z_{i}^{0}$ and $z_{i}^{1}$ are connected through a residual connection, serving as the input to the Transformer module. Within the Transformer module, the Multi-Head Attention layer and Add&Norm layers are first utilized for learning. Then, the Feed-Forward layer, composed of fully connected layers and ReLU activation functions, along with the subsequent Add&Norm layers, introduce nonlinearity into the module. This approach aims to enhance the learning process, ultimately yielding the output of the Transformer learner. The specific computation steps are shown in Equations (15)–(17):

$$z_{i}^{1} = LSTM\left(z_{i}^{0}\right) \tag{15}$$

$$z_{i}^{2} = Fusion\left(z_{i}^{0} + z_{i}^{1}\right) \tag{16}$$

$$z_i^3 = Transformer\left(z_i^2\right) \tag{17}$$

The output $z_i^3$ from the Transformer block learner serves as the final relation representation for the task triplet $(h,r,t)$, denoted as $z(h,r,t)$. In the end, for each few-shot relation $r$ and its corresponding task triplet $(h,r,t) \in S_r / Q_r$, there exists a corresponding final relation representation.

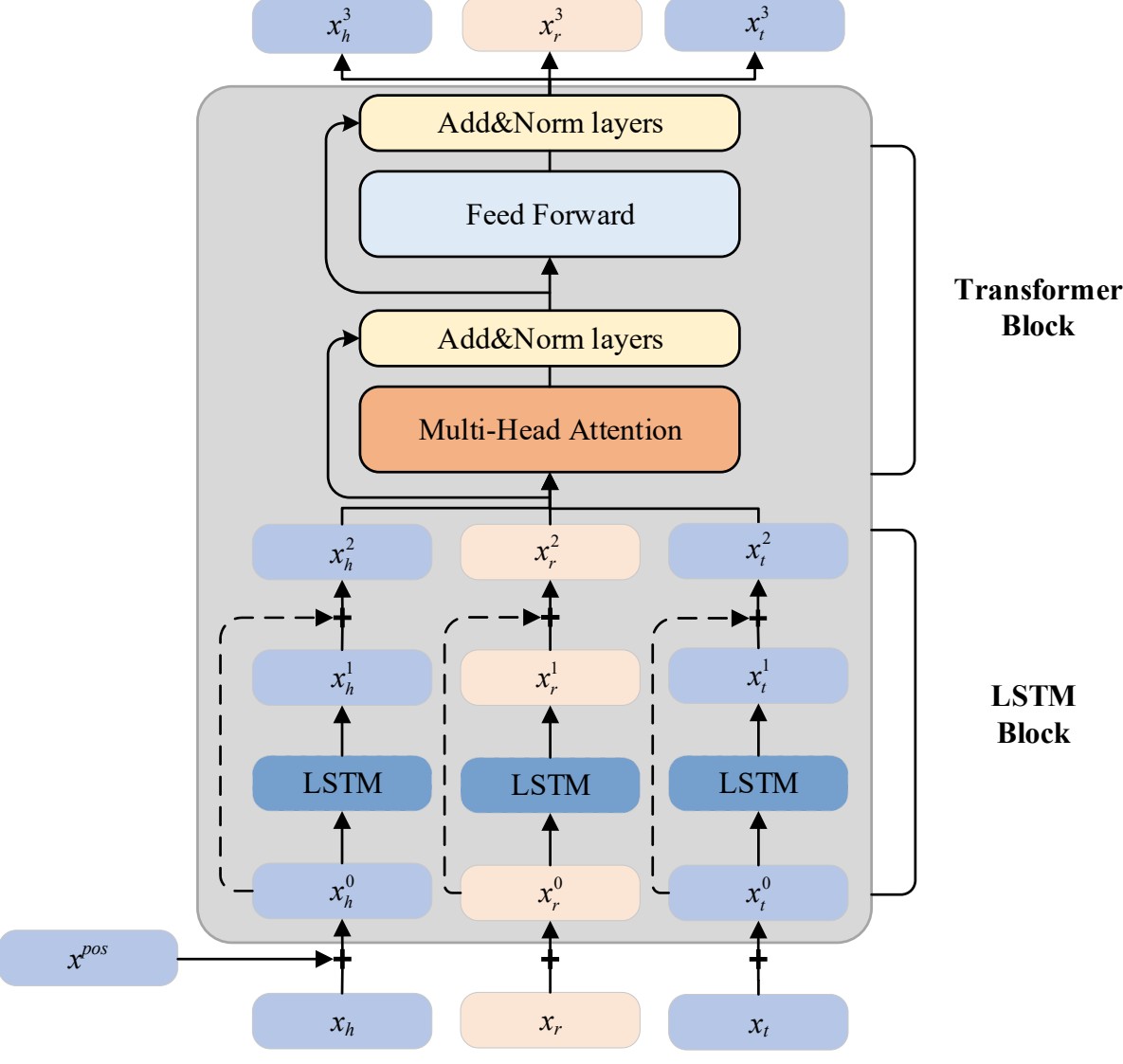

**Figure 5.** The main structure of a learning framework composed of a Transformer and an LSTM.

### 4.4. Matching Process Computation

After being processed by the Transformer learner, each entity in the reference set and query set obtains its corresponding relation representation. Similar to previous methods 20, FRL-KGC calculates the semantic similarity between the query triplet and the reference set using a metric-based approach, and selects the triplet with the highest similarity score as the model's prediction. The main structure of the matching process computation is illustrated in Figure 6.

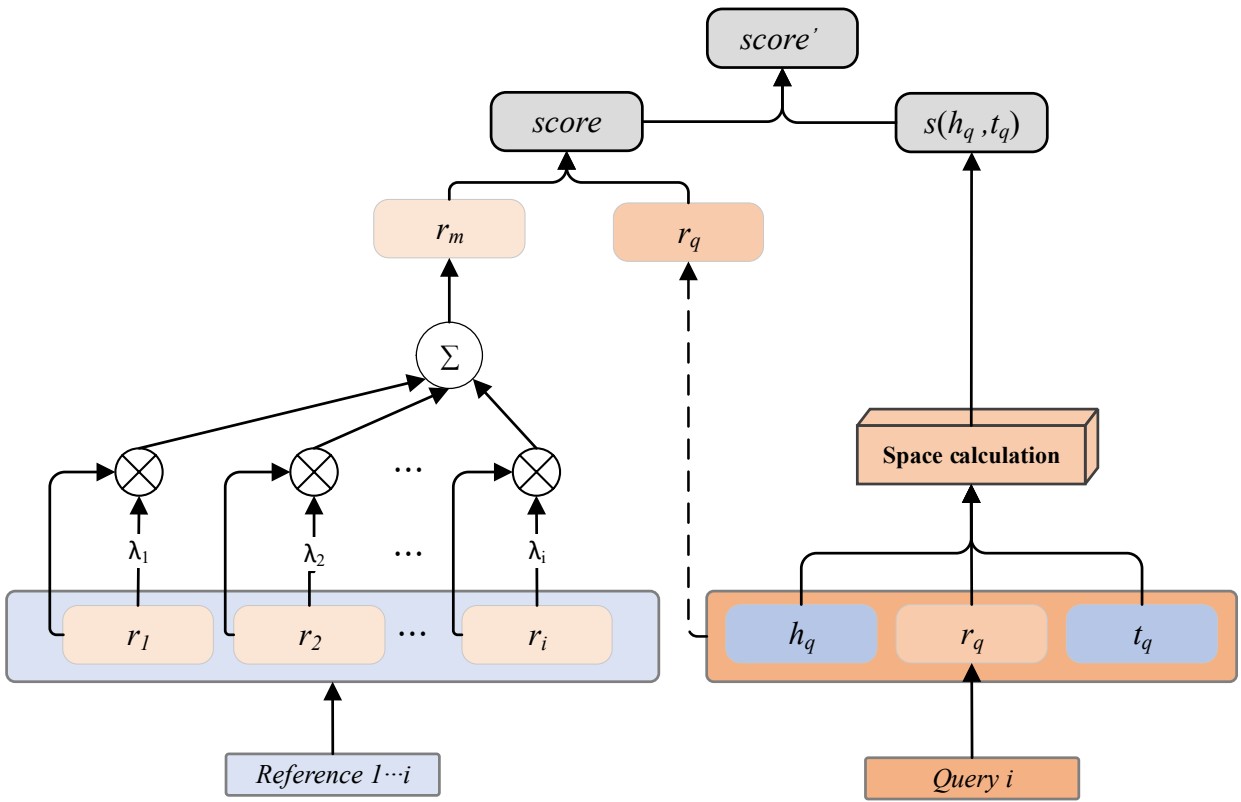

**Figure 6.** The schematic diagram of matching process calculation.

However, during the matching computation, each reference set contains different information 30, and it is important to dynamically learn the representation of the reference relation $r$ for different query triplets. Inspired by the work in reference 31, FRL-KGC generates a dynamic query relation representation $\boldsymbol{r_m}$ for each query entity pair. The computation process is shown in Equations (18) and (19):

$$\boldsymbol{r_m} = \sum\nolimits_{\boldsymbol{S_{r_i}} \in \mathcal{S}_r} \lambda_i \cdot \boldsymbol{S_{r_i}} \tag{18}$$

$$\lambda_i = softmax(\cos(\boldsymbol{S_{r_i}}, q_r)) = \frac{\exp(\cos(\boldsymbol{S_{r_i}}, q_r))}{\sum\nolimits_{S_{r_j} \in \mathcal{S}_r} \exp(\cos(S_{r_j}, q_r))} \tag{19}$$

where $\lambda_i$ denotes the $softmax(\cdot)$ attention weights for the representation of each reference relation; $\boldsymbol{S_{r_i}}$ represents the relation representation of the *i-th* entity pair in the reference set obtained through the Transformer learner (Equation (18)); $q_r$ represents the relation representation of the query entity pair obtained through the Transformer learner (Equation (18)); $\cos(\cdot)$ represents the cosine similarity; and $j$ represents the size of the reference set.

Next, the dynamic query relation representation $\boldsymbol{r_m}$ and the semantic similarity $score(\boldsymbol{r_m}, q_r)$ between $\boldsymbol{r_m}$ and the query $q_r$ are computed as shown in Equation (20):

$$score(\boldsymbol{r_m}, q_r) = \boldsymbol{r_m} \cdot q_r \tag{20}$$

where the higher the value of $score(\cdot)$, the greater the semantic similarity between the query entity pair $(h, t)$ under the reference relation $r$ and the few-shot reference set,

indicating a higher possibility of the query triplet being valid. Conversely, a lower $score(\cdot)$ suggests a smaller possibility.

However, in translational models like TransE 10, it is important to consider the translational property for the expectation of $h + r \approx t$. According to Equation (18), $\left(z_h^3, z_r^3, z_t^3\right)$ represents the output sequence of the query entity pair $\left(h_q, t_q\right)$ after passing through the Transformer learner. The translation score $s$ is defined as shown in Equation (21):

$$s\left(h_q, t_q\right) = \left\| z_h^3 + z_r^3 - z_t^3 \right\|_2 \tag{21}$$

where $\left\| z_i^3 \right\|_2$ represents the $L_2$ norm of vector $z_i^3$, and $s$ represents the distance between $z_h^3 + z_r^3$ and $z_t^3$. Therefore, the smaller the value of $s$, the higher the possibility of the query triplet being valid.

Therefore, considering both the values of $score(\cdot)$ and $s(\cdot)$, the calculation of the final matching $score'(\cdot)$ for the query entity pair $\left(h_q, t_q\right)$ is determined as shown in Equation (22):

$$score'\left(h_q, t_q\right) = score\left(\boldsymbol{r_m}, q_r\right) - \delta s\left(h_q, t_q\right) \tag{22}$$

where $\delta$ is an adjustment factor. The final $score'(\cdot)$ is calculated for all the query triplets and sorted accordingly.

### 4.5. Loss Function

This paper follows the model training settings of other FKGC methods 16. Given a task relation $r \in \mathcal{R}_{train}$ and its corresponding triplets, a reference set $\mathcal{S}_r = \{(h_i, t_i) | (h_i, r, t_i) \in \mathcal{G}\}_i$ is constructed by randomly sampling $K$ triplets from the triplet set, while B (batch_size) triplets are randomly sampled from the remaining triplets as the positive query set $\mathcal{Q}_r = \left\{\left(h_q, t_q\right) | \left(h_q, r, t_q\right) \in \mathcal{G}\right\}$. As there are no negative triplets in the knowledge graph itself, a corresponding negative query set $\mathcal{Q}_r^- = \left\{\left(h_q, t_q^-\right) | \left(h_q, r, t_q\right) \in \mathcal{G}, \left(h_q, r, t_q^-\right) \notin \mathcal{G}\right\}$ is constructed by replacing the tail entity of the triplets in $\mathcal{Q}_r$. The hinge loss function is used for training, as shown in Equation (23):

$$\mathcal{L} = \sum_r \sum_{\left(h_q, t_q\right) \in \mathcal{Q}_r} \sum_{\left(h_q, t_q^-\right) \in \mathcal{Q}_r^-} [\gamma + score'\left(h_q, t_q^-\right) - score'\left(h_q, t_q\right)]^+ \tag{23}$$

where $[\cdot]^+ = max(0, x)$ represents the standard hinge loss function, and $\gamma$ is a margin hyperparameter used to separate positive and negative query triplets.

Finally, this paper adopts the batch sampling-based meta training strategy proposed in Reference 17, which minimizes $\mathcal{L}$ while optimizing the model parameter set. $L_2$ regularization is applied to the model parameter set to avoid overfitting, and the Adam optimizer 32 is used to optimize the model.

In summary, the training process of the model is shown in Algorithm 1.

---

**Algorithm 1** The Training Process of FRL-KGC Model

---

**Input:** Training Task Set $T_{meta-training}$, TransE knowledge graph embedding vector, Initialization parameter $\theta$ of matrix model, Reference sample size;

**Output:** Optimization parameters of the model $W_C$, $\lambda_c$, $\theta$

1   For epoch in 1 to M do

2       Shuffle($T_{meta-training}$)   // Disrupt tasks in $T_{meta-training}$

3       For $\mathcal{T}_r$ in $T_{meta-training}$ do

4           $\mathcal{S}_r = Sample(r, K, \mathcal{G})$   // Extract entity pairs of relation $r$ from $K$ $\mathcal{G}$ as a small sample reference set $\mathcal{S}_r$

5           For $k$ in $K$ do

6               Enhance the embedding vector representation of head and tail entities, and update the representation of few-show relations.

7           End For

8           Process triples through Transformer learners

9           $\mathcal{Q}_r = Sample(r, \mathcal{G}) - \mathcal{S}_r$   // Build a regular triplet query set $\mathcal{Q}_r$

10          $\mathcal{Q}_r^- = Pollute(\mathcal{Q}_r)$   // Pollute the tail entity of a positive triplet to obtain a negative triplet

11          Calculate matching scores

12          Accumulate the batch loss $\mathcal{L}$

13          Update $\theta$   // Update $\theta$ using the Adam optimizer

14      End for

15  End For

---

## 5. Experiments

In this section, link prediction experiments are conducted on the NELL-One and Wiki-One datasets, which are constructed with reference to 16, and the FRL-KGC model is compared with six few-shot knowledge graph completion models and five traditional models. Meanwhile, the performance of the models is evaluated on the basis of MRR, Hits@10, Hits@5 and Hits@1.

### 5.1. Datasets and Evaluation Indicators

The two benchmark datasets used in this paper, NELL-One and Wiki-One, are commonly used datasets for few-shot knowledge graph completion tasks. NELL-One is constructed using the NELL dataset 4, which is a system that extracts structured information from web text and automatically expands and extends the knowledge graph. Wiki-One is a subset extracted from the Wikidata knowledge base 5. In both datasets, relations with more than 50 but fewer than 500 associated triples are defined as few-shot relations (task relation $\mathcal{R}_{task}$), while the remaining relations related to these triples constitute the background knowledge graph $\mathcal{G}'$. Following the FKGC task setup, the task relations in NELL-One and Wiki-One are divided into training relations $\mathcal{R}_{train}$, validation relations $\mathcal{R}_{validation}$, and test relations $\mathcal{R}_{test}$ in the proportions of 51/5/11 and 133/16/34, respectively. Detailed statistics of the datasets are presented in Table 2.

**Table 2.** Statistics of datasets. # Ent. denotes the number of unique entities. # Rel. denotes the number of few-shot relations. $\mathcal{R}_{task}$ denotes the number of relations we use as few-shot tasks.

| Dataset | # Ent. | # Rel. | # Triples | $\mathcal{R}_{task}$ | $\mathcal{R}_{train}$ | $\mathcal{R}_{validation}$ | $\mathcal{R}_{test}$ |
|---------|--------|--------|-----------|----------|-----------|----------------|----------|
| NELL-One | 68,545 | 58 | 181,109 | 67 | 51 | 5 | 11 |
| Wiki-One | 4,838,244 | 22 | 5,859,240 | 183 | 133 | 16 | 34 |

In this study, the Mean Reciprocal Rank (MRR) and Hits@n are used as evaluation metrics. MRR calculates the average of the reciprocals of the ranks of the correct answers. Higher MRR values indicate better model performance. Hits@n measures the proportion of correct answers appearing in the top n ranks. Similarly, higher Hits@n values indicate better model performance. In this experiment, *n* is set to 1, 5, and 10. Moreover, the size of the reference set (*K*) is set to 5, which means all models are evaluated on a five-shot knowledge graph completion task.

*5.2. Baseline Methods*

To evaluate the effectiveness of the FRL-KGC model, two categories of benchmark models are selected for comparison: traditional knowledge graph embedding methods and FKGC methods.

For the traditional knowledge graph embedding models, five models are chosen as control models: TransE 10, DistMult 12, ComplEx 13, SimplE 33, and RotatE 34. During the training process of traditional knowledge graph embedding models, all triples from the background knowledge graph $\mathcal{G}'$ and the task relation set $\mathcal{R}_{task}$ are used for training. The task relation set $\mathcal{R}_{task}$ also includes the reference triples used in the validation and testing phases of the FKGC task.

The FKGC models chosen for comparison in this study include GMatching 16, MetaR 19, FSRL 17, FAAN 18, and GANA 20. Among them, GMatching is compared using GMatching (MaxP), which includes a neighborhood encoder and a matching processor and performs few-shot reasoning tasks through max pooling. MetaR is divided into two scenarios: MetaR (Pre-train) and MetaR (In-train). MetaR (Pre-train) trains entity embeddings using only the background knowledge graph, while MetaR (In-train) samples triples from the background knowledge graph and the original training set and includes them in the model training process.

In the comparative experiments, both traditional knowledge graph embedding models and few-shot knowledge graph completion models were evaluated using their respective optimal parameter settings. The five-shot knowledge graph completion experiments were conducted five times, and the average of the results was taken as the final result. This approach ensures a fair comparison and provides a more reliable evaluation of the models' performance.

*5.3. Implementation Details*

The FRL-KGC model is implemented using the PyTorch framework and the experiments are conducted on a single NVIDIA GeForce RTX 4090 GPU. The pretrained embedding model selected is the TransE model 10. The relevant parameters for training on NELL-One and Wiki-One are as follows: entity and relation embedding dimensions are set to 100 and 50, respectively; batch size is set to 128; the initial learning rates ($lr$) are set to $5 \times 10^{-5}$ and $6 \times 10^{-5}$ for NELL-One and Wiki-One, respectively; the maximum number of neighbors (M) is fixed at 150; the hyperparameter $\gamma$ is set to 5; and the Adam optimizer is used. In the first 10,000 steps of the training process, the model gradually increases the learning rate and then linearly decreases it. Model validation is performed every 10,000 training steps, and the maximum number of training steps is set to 300,000.

During the model validation process, the model parameters with the highest MRR value are selected as the optimal training result for the FRL-KGC model.

*5.4. Results*

The five-shot link prediction results of all models on the NELL-One and Wiki-One datasets are shown in Table 3. It can be observed from Table 3 that:

(1) Compared with traditional knowledge graph embedding methods, FRL-KGC achieves the best performance on both datasets. The experimental results demonstrate that FRL-KGC can effectively predict missing entities in few-shot relations.

(2) On both datasets, the FRL-KGC model outperforms the best results of the baseline models on four evaluation metrics. Compared with the best-performing MetaR (In-train) model on the NELL-One dataset, the FRL-KGC model improves the MRR, Hits@10, Hits@5, and Hits@1 metrics by 2.9%, 1.9%, 3.1%, and 4.3%, respectively. The performance improvements on the Wiki-One dataset are 3.3%, 4.3%, 3.4%, and 3.2%, respectively. It is worth noting that only one setting in either Pre-train or In-train performs well on a single dataset. This indicates that our model has better generalization ability across different datasets. Furthermore, FRL-KGC can leverage the contextual semantics and structural information of entities in KG to improve the performance of few-shot knowledge graph completion.

**Table 3.** Results of five-link prediction on NELL-One and Wiki-One. Bold numbers denote the best results, and underlined numbers indicate suboptimal results.

| Model | NELL-One | | | | Wiki-One | | | |
|---|---|---|---|---|---|---|---|---|
| | MRR | Hits@10 | Hits@5 | Hits@1 | MRR | Hits@10 | Hits@5 | Hits@1 |
| Traditional models | | | | | | | | |
| TransE | 0.176 | 0.316 | 0.234 | 0.109 | 0.134 | 0.188 | 0.158 | 0.106 |
| DistMult | 0.211 | 0.312 | 0.256 | 0.135 | 0.076 | 0.154 | 0.101 | 0.024 |
| ComplEx | 0.186 | 0.299 | 0.231 | 0.119 | 0.081 | 0.182 | 0.121 | 0.032 |
| SimplE | 0.156 | 0.284 | 0.225 | 0.094 | 0.097 | 0.181 | 0.125 | 0.045 |
| RotatE | 0.176 | 0.331 | 0.245 | 0.109 | 0.052 | 0.091 | 0.065 | 0.026 |
| Few-shot models | | | | | | | | |
| GMatching (MaxP) | 0.176 | 0.294 | 0.233 | 0.113 | 0.263 | 0.387 | 0.337 | 0.197 |
| MetaR (Pre-train) | 0.162 | 0.282 | 0.233 | 0.101 | 0.320 | 0.443 | <u>0.397</u> | 0.262 |
| MetaR (In-train) | <u>0.308</u> | <u>0.502</u> | <u>0.423</u> | <u>0.210</u> | 0.229 | 0.323 | 0.289 | 0.197 |
| FSRL | 0.269 | 0.482 | 0.369 | 0.178 | 0.221 | 0.269 | 0.183 | 0.163 |
| FAAN | 0.265 | 0.416 | 0.347 | 0.187 | 0.314 | <u>0.451</u> | 0.384 | 0.245 |
| GANA | 0.296 | 0.497 | 0.412 | 0.194 | <u>0.324</u> | 0.437 | 0.375 | <u>0.264</u> |
| FRL-KGC (ours) | **0.337** | **0.521** | **0.454** | **0.253** | **0.357** | **0.494** | **0.431** | **0.296** |

*5.5. Ablation Study*

The framework of the FRL-KGC model consists of three key components: (a) a high-order neighborhood entity encoder based on a gating mechanism; (b) a relation representation encoder; and (c) a Transformer learner. To assess the impact of each component on the overall performance of FRL-KGC, in this paper, ablation experiments are conducted on the Wiki-One dataset for five-shot link prediction.

(a) To investigate the effectiveness of the high-order neighborhood entity encoder based on a gating mechanism, modifications are made as follows: A1_a encodes only first-order neighborhood entities for output; A1_b removes the gating mechanism and uses the average embedding of neighborhood entities instead of $c_h$.

(b) To study the effectiveness of the relation representation encoder, modifications are made as follows: A2_a simply uses the average embedding of the reference entity pairs as the representation of the relation.

(c) To examine the effectiveness of the Transformer learner, modifications are made as follows: A3_a removes the LSTM module; A3_b removes the Transformer module.

Table 4 presents the comparative experimental results after removing each component.

**Table 4.** Ablation study of FRL-KGC under five-shot settings on Wiki-One. Bold numbers denote the best.

| Ablation on Model | five-shot on Wiki-One | | | |
| :---: | :---: | :---: | :---: | :---: |
| | MRR | Hits@10 | Hits@5 | Hits@1 |
| A1_a | 0.314 | 0.443 | 0.386 | 0.258 |
| A1_b | 0.336 | 0.469 | 0.395 | 0.272 |
| A2_a | 0.343 | 0.483 | 0.425 | 0.267 |
| A3_a | 0.331 | 0.453 | 0.383 | 0.279 |
| A3_b | 0.301 | 0.432 | 0.371 | 0.264 |
| FRL-KGC (ours) | **0.357** | **0.494** | **0.431** | **0.296** |

The results in Table 4 demonstrate that the performance of the complete FRL-KGC model outperforms all of its variants. This indicates that: (a) the high-order neighborhood entity encoder based on the gate mechanism can effectively enhance the information contained in the center entity, and the gate mechanism can filter out the impact of noisy neighbors; (b) by introducing the neighborhood relation representation of entity pairs in the reference set, FRL-KGC can improve the quality of relation embeddings, facilitate relation prediction, and reduce dependence on entity pairs, thus improving the model's generalization ability; (c) the combined structure of the LSTM network and the Transformer module in the Transformer learner is better than using them separately, where the LSTM network can effectively enhance fine grained contextual semantic representation, and combined with the powerful Transformer module can improve the accuracy of link prediction.

*5.6. Impact of Few-Shot Size*

To illustrate the impact of the few-shot size on the model's performance, impact of few-shot size experiments were conducted on the Wiki-One dataset, as shown in Figure 7. The horizontal axis represents the size $K$ of the reference set, and the vertical axis represents the evaluation index.

As shown in Figure 7, as the number of instances in the reference set $K$ increases, the MRR and Hits@1 values of FRL-KGC gradually increase, and the overall prediction accuracy of all models improves. However, after reaching a certain level, the performance improvement tends to flatten and then decreases again. This indicates that the size of the reference set is a crucial factor affecting the accuracy of link prediction when predicting new triplets. With a larger reference set, there is more reference information for the query triplet, which can improve the accuracy of link prediction. However, when the reference set is too large, the prediction accuracy may decrease. This is due to the fact that the model may learn more irrelevant information when learning the relation representation with more reference information, leading to a decrease in prediction accuracy. Additionally, with more reference information, entities may have more meanings, which increases the complexity of relation learning, leading to a decrease in prediction accuracy.

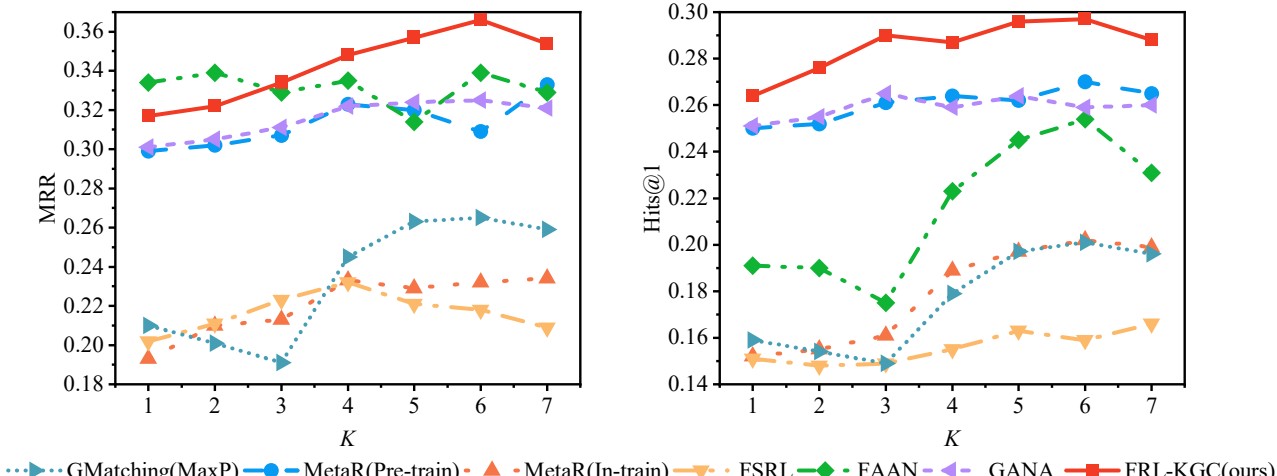

**Figure 7.** Impact of few-shot size *K* in the performance of FKGC methods on Wiki-One dataset.

## 6. Conclusions and Future Work

This study proposes a Few-shot Relation Learning-based Knowledge Graph Completion model (FRL-KGC), specifically designed for few-shot knowledge graph completion tasks. FRL-KGC incorporates a gating mechanism during the aggregation of high-order neighborhood entity information, effectively filtering out noise from neighboring entities and improving the quality of entity encoding. In the process of learning relation representations, FRL-KGC leverages the information embedded in the neighborhood relations of entity pairs in the reference set, enhancing the quality of relation embeddings and reducing reliance on specific entity pairs, thus improving the model's generalization ability. Furthermore, the introduction of an LSTM network in the Transformer learner further improves the quality of few-shot relations. The experimental results indicate that the FRL-KGC model outperforms existing FKGC models in terms of link prediction accuracy. However, the design of the dataset does not fully capture the dynamic nature of few-shot knowledge graphs. If the knowledge graph undergoes real-time changes, ensuring the model's inference accuracy becomes a challenging problem. In our future work, we plan to explore the use of timestamps to enhance the model's representation capacity of the knowledge graph and maintain inference accuracy in dynamic few-shot knowledge graph learning. Additionally, we will investigate the use of external knowledge sources to augment the representations of entities and relations, such as leveraging textual descriptions of entities and relations.

**Author Contributions:** Conceptualization, J.G. and W.L.; Methodology, J.G. and W.L.; Software, J.G.; Validation, A.L., Y.G. and X.Z.; Formal analysis, A.L., Y.G. and X.Z.; Resources, Y.G.; Data curation, J.G.; Writing—original draft, J.G.; Writing—review & editing, W.L. and A.L.; Visualization, A.L.; Supervision, Y.G. and X.Z.; Project administration, W.L.; Funding acquisition, W.L. All authors have read and agreed to the published version of the manuscript.

**Funding:** This work was supported by the Basic Scientific Research in Central Universities of North Minzu University (2021JCYJ12), the Ningxia Natural Science Foundation Project (2021AAC03215), the National Natural Science Foundation of China (62066038, 61962001).

**Institutional Review Board Statement:** Not applicable.

**Informed Consent Statement:** Not applicable.

**Data Availability Statement:** The data presented in this study are available on request from the corresponding author.

**Conflicts of Interest:** The authors declare no conflict of interest.

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
