# Peer review of "Few-Shot Knowledge Graph Completion Model Based on Relation Learning"

_applsci, doi:10.3390/app13179513_

Round 1
Reviewer 1 Report
This manuscript presents a few-shot model knowledge graph completion. Generally this text follows extremely close to the structure and language from [17]. This is very apparent in the introduction and examples.
The actual novelty of this manuscript is left as an exercise to the reader. The introduction provides a brief glimpse at the shortcomings to the state of the art, and the section 3 provides descriptions but no critical comparison.
In particular, the exact differences and advances, given the extreme similarity in topic and structure, to [17] are not expounded. The fact that it's not is particularly problematic to the point that it's difficult to actually parse any scientific novelty and provide a reasonable recommendation.
The manuscript should receive a thorough review of the English language. There are many simple typos (e.g., meat-test), missing articles before nouns (e.g., in the abstract), and weird hyphenations in the abstract.
Author Response
Response to Reviewer 1 Comments
Point 1: This manuscript presents a few-shot model knowledge graph completion. Generally this text follows extremely close to the structure and language from [17]. This is very apparent in the introduction and examples.
Response 1: The structure of this paper may have some similarities with the literature [17], but the content is completely different. In the introduction, this paper provides a summary of the four parts of the model and briefly describes the improvements compared to other models. In contrast, literature [17] only provides a simple comparison with the GMatching model. To address the shortcomings of other models, we made modifications in the introduction.
For example: "In the FKGC task, the aforementioned models have achieved good results. However, these methods still have some limitations:1) FAAN [18] fails to effectively utilize the valuable information from high-order neighboring entities (the most relevant high-order neighborhood set in non-local graphs[21]) and cannot differentiate the importance of different neighbor information, leading to noise-related issues.2) FSRL[17] simply uses a recurrent autoencoder to aggregate a small reference set. However, as training progresses, the FSRL model becomes overly reliant on entity embeddings, resulting in relationship overfitting and a decrease in model generalization capability.3) FSRL does not consider the translational property of the TransE model during matching queries, which can lead to a decline in matching accuracy.4) Previous methods have not adequately considered the impact of entity pairs on contextual semantics[21], resulting in reduced accuracy in link prediction"
In the example section, both the structure and content of this paper are completely different from literature [17]. In terms of structure, literature [17] only provides a simple introduction to few-shot learning and KG relation learning, and the experiments are also limited to these two aspects. In contrast, this paper introduces both traditional KG completion models and few-shot KG completion models, which is entirely different. We cover a broader range of aspects.
Point 2: The actual novelty of this manuscript is left as an exercise to the reader. The introduction provides a brief glimpse at the shortcomings to the state of the art, and the section 3 provides descriptions but no critical comparison.
Response 2: Thank you for providing the details about the innovations mentioned in the introduction of your paper.
To improve the accuracy of link prediction, this paper proposes a few-shot knowledge graph completion model called FRL-KGC, which makes the following contributions:
(1) We introduce the FRL-KGC model, which incorporates a gating mechanism to extract valuable contextual semantics from the head entity, tail entity, and neighborhood information, specifically addressing high-order neighborhood information in the knowledge graph. Furthermore, we leverage the correlation between entity pairs in the reference set to represent relationships, reducing the dependency of relationship embeddings on the central entity.
(2) We effectively utilize both the structural and textual information of the knowledge graph to capture features related to few-shot relations.
(3) Experimental evaluations are conducted on two publicly available datasets, demonstrating that our proposed model outperforms other mainstream KGC methods. Additionally, ablation experiments validate the effectiveness of each key module in our model.
Regarding the issue of critical comparison, it has been addressed in the previous response.
Point 3: In particular, the exact differences and advances, given the extreme similarity in topic and structure, to [17] are not expounded. The fact that it's not is particularly problematic to the point that it's difficult to actually parse any scientific novelty and provide a reasonable recommendation.
Response 3: This paper and the reference [17] share a similar topic as they address different approaches to solve the same problem. However, the two papers are distinct in terms of their content. In [17], the heterogeneous neighborhood encoder utilizes an attention module, whereas in this paper, we employ a gating mechanism to differentiate the importance of information contained in different neighbors. Furthermore, we introduce improved Transformer modules and a matching computation module in the overall structure of our model. Therefore, this paper is completely different from the content presented in [17].
In the experimental section, we conducted comparative experiments on 5-shot link prediction using six few-shot knowledge graph completion models and five traditional knowledge graph completion models on publicly available NELL-One and Wiki-One datasets. FRL-KGC outperformed all the comparison models in terms of MRR, Hits@10, Hits@5, and Hits@1 metrics. Additionally, a detailed analysis of our model was presented in Section 5.5, Ablation Study. In Section 5.6, Impact of Few-Shot Size, we included a comparative experiment with the GMatching(MaxP) model and updated Figure 7 while providing a description of the experiments.
Point 4: The manuscript should receive a thorough review of the English language. There are many simple typos (e.g., meat-test), missing articles before nouns (e.g., in the abstract), and weird hyphenations in the abstract.
Response 4: We have conducted a comprehensive English check of the manuscript. We have corrected spelling errors and removed unnecessary hyphens. We have corrected the terminology issue. For example, relationship/relationship, set/collection, task/problem, etc.

Reviewer 2 Report
Dear Authors
Thank you for the interesting work. There are some of points which will modify the paper.
Regards,

Dear Authors
Please, try to check the proofreading.
Regards,
Author Response
Response to Reviewer 2 Comments
Point 1: The advantages of this study must be added in the introduction
Response 1: We have added critical comparisons and descriptions in the introduction to highlight the advantages and innovations of this paper. The revised paragraph is as follows:
" In the FKGC task, the aforementioned models have achieved good results. However, these methods still have some limitations: 1) FAAN fails to effectively utilize the valuable information from high order neighboring entities (the most relevant high order neighbor-hood set in nonlocal graphs[21]) and cannot differentiate the importance of different neighbor information, leading to noise related issues. 2) FSRL simply uses a recurrent au-toencoder to aggregate a small reference set. However, during the training process, he FSRL model tends to excessively depend on entity embeddings, leading to overfitting of relations and a decline in the generalization capability of the model. 3) FSRL does not consider the translational property of the TransE model during matching queries, which can lead to a decline in matching accuracy. 4) Previous models have not adequately con-sidered the impact of entity pairs on contextual semantics[21], resulting in reduced accu-racy in link prediction."
Point 2: Authors should enhance the introduction section by adding some more recent articles in the study.
Faheem, M, Arefin, M.A,Khalifa, H.A. Zahid, Z,and Siddique, I (2022).Computing fault- tolerant resolvability for the families of kayak paddles anddragon graphs.Journal of Mathematics, Vol.2022,Article ID 6873039.9pages.
Response 2: We were unable to find the references you provided in the Journal of Mathematics journal or Google Scholar. Please verify the accuracy of the provided references to facilitate our citation and enrich the content of the article.
Point 3: Author should state the objective of the work in the bullet point.
Response 3: We have provided a more detailed explanation of the key points and objectives of our proposed model in the manuscript. The revised paragraph is as follows:
To improve the accuracy of link prediction, this paper proposes a few-shot knowledge graph completion model (FRL-KGC), which makes the following contribu-tions:
(1) We introduce the FRL-KGC model, which incorporates a gating mechanism to ex-tract valuable contextual semantics from the head entity, tail entity, and neighborhood in-formation, specifically addressing high order neighborhood information in the knowledge graph. Furthermore, we leverage the correlation between entity pairs in the reference set to represent relations, reducing the dependency of relation embeddings on the central entity.
(2) We effectively utilize both the structural and textual information of the knowledge graph to capture features related to few-shot relations.
(3) Experimental evaluations are conducted on 2 publicly available datasets, and the results demonstrate that our proposed model outperforms other KGC models. Addition-ally, ablation experiments validate the effectiveness of each key module in our model.
Point 4: Authors should check all equation, some are not correct in symbols.
Response 4: We have verified all the equations and corrected any incorrect symbols.
Point 5: The comparison with the existing method must be done
Response 5: In the experimental section, we compared our model with existing baseline models. The existing models can be categorized into two main types: traditional knowledge graph completion models and few-shot knowledge graph completion models. Additionally, we conducted new experiments specifically focusing on traditional knowledge graph completion methods. The experimental conclusion is as follows:
The 5-shot link prediction results of all models on the NELL-One and Wiki-One da-tasets are shown in Table 3. It can be observed from Table 3 that:
1) Compared with traditional knowledge graph embedding methods, FRL-KGC achieves the best performance on both datasets. The experimental results demonstrate that FRL-KGC can effectively predict missing entities in few-shot relations.
2) On both datasets, the FRL-KGC model outperforms the best results of the baseline models on 4 evaluation metrics. Compared with the best-performing MetaR (In-train) model on the NELL-One dataset, the FRL-KGC model improves the MRR, Hits@10, Hits@5, and Hits@1 metrics by 2.9%, 1.9%, 3.1%, and 4.3%, respectively. The performance im-provements on the Wiki-One dataset are 3.3%, 4.3%, 3.4%, and 3.2%, respectively. It is worth noting that only one setting in either Pre-train or In-train can perform well on a sin-gle dataset. This indicates that our model has better generalization ability across different datasets. Furthermore, FRL-KGC can leverage the contextual semantics and structural in-formation of entities in KGs to improve the performance of few-shot knowledge graph completion.
In Section 5.6, Impact of Few-Shot Size, we included a comparative experiment with the GMatching(MaxP) model and updated Figure 7 while providing a description of the experiments.
Point 6: The limitation must be done.
Response 6: We have added some content in the conclusion to reflect the limitations of this study.
The experimental results indicate that FRL-KGC outperforms the state-of-the-art FKGC model in terms of link prediction accuracy. However, the design of the dataset does not fully reflect the dynamic nature of few-shot knowledge graphs. If the knowledge graph is subject to real-time changes, ensuring the inference accuracy of the model becomes a challenging problem. We consider this as a direction for future work, aiming to address the issue of maintaining inference accuracy in dynamic and evolving few-shot knowledge graphs.
Point 7: Future research work should be added in the conclusion section.
Response 7: In our future work, we plan to explore the incorporation of timestamps to enhance the representational capacity of the knowledge graph. Additionally, we will investigate the use of external knowledge sources to augment the representations of entities and relations, such as leveraging textual descriptions of entities and relations. These approaches aim to improve the inference capabilities of the FKGC model and enhance its overall reasoning abilities.
Point 8:Proofreading must do.
Response 8: We have conducted a comprehensive English check of the manuscript. We have corrected spelling errors and removed unnecessary hyphens. We have corrected the terminology issue. For example, relationship/relationship, set/collection, task/problem, etc.

Reviewer 3 Report
The paper belongs to the area of knowledge graph enrichment applying machine learning methods and fits the topic of special issue well. The concrete problem considered is narrower: to enrich so called “few-shot” relations (50-500 instances in a knowledge graph). However, many works investigating the problem have been published during last years.
Positive points of the paper are that the authors investigate the existing ideas and techniques, combine and improve them further, and propose yet another Knowledge Graph Completion model based on few-shot relation learning. According to the experimental results, the final model outperforms the competitors. Additional experiments show that all main components of the model are significant for the complete model. The impact of the few-shot size on the model's performance is also investigated.
Regretfully, the paper contains MANY unclear points listed below (I am not sure that I caught all of them). Definitely, all these points should be clarified and revised. The whole style of presentation should be clarified and improved. In the present form sections 3 and 4 are very difficult to follow.
Another pack of issues concern experimental part:
· Experiments for 6 models from 11 are not reproduced, results are just taken from [34].
· Implementation details of other 5 models are not mentioned. Do the respective authors provide source code of their models?
· The authors do not provide a way to reproduce their results (probably the experimental framework should be published).
Unclear points discovered:
· G’ is defined as G with removed “all subgraphs of task relationships” in Table 1 and “by removing all task relationships” in 3.2. So subgraphs or just relations?
· [Table 1] h, r, t denotes both triplets end embeddings
· [Table 1] (h, t) in description is not related to R’r
· [lines 203-204] “The candidate tail entities for each query triplet are constructed based on entity type constraints[18]” – should be clarified.
· [Section 4.1]
o No explicit definition of “i-th higher-order neighbor relationship” is provided. An example should be also provided.
o Definition of “softmax” function is not referenced explicitly.
o Correspondence of h and h0 is not clear.
o et, and eh are not explained
· [Fugure 3]
o “Selective attention” is not commented
o GLU should be put instead of “GRU”
o GLU block is neither explained nor referenced. I can not find Ct-1, Ct, xt variables from the figure in the text
o eh and ch are used to produce e on the figure. But formula (6) uses h, but not eh.
· [line 322] Formula should be revised somehow. In this form it is syntactically incorrect.
· [line 324] What is Er?
· [Fugure 5] x variables with different indices are used on the figure, but z variables are used in the text.
· [line 383] “Fusion” is not defined.
· [line 425] ||z3i||2 represents the norm of z3i, not z
· [Table 2] It seems that #Rel is a number of non-few-shot relations, and not the number of all relations as defined in the caption
Remarks:
· I doubt that exact percentage of outperforming should be included in the abstract.
· “Relation” and “relationship” is used in an interchangeable way. It seems only “relation” should be used. Even in the title of the paper and in the title of the proposed model different terms are used.
· Several existing works apply few-shot relation learning for Knowledge Graph Completion. For instance, in [16][20] the same terms are used even in titles. The authors should provide more specific title for their model to distinguish it from the others.
· [Introduction] Texts in [69-80] lines and in [81-90] lines are very similar. The pieces should be combined, duplications should be removed.
· [Introduction, 60-68] “the high-order neighborhood entity information” and “the influence of entity pairs on the contextual semantics” are not understandable within the Introduction. Additional explanations and references are required.
· [Section 2.1] “Translation-based methods treat relations as translation operations between entity pairs” – should be explained.
· [Section 2.3] Methods from this section are not presented in experiments. It should be explained why.
· [Section 3] Table 1 does not help much if placed in this section. Terminology in description is somehow understandable only after reading the respective sections of the paper.
· [lines 199-200] “The associated triplets are randomly divided into a reference set Sr, a query set Qr, and a set Dr={Sr, Qr}”: triplets are divided into Sr and Qr. Dr is just a pair of these sets.
Minor remarks:
· The text of the whole paper should be carefully checked for redundant hyphens like “con-tained” or “en-tities” in abstract.
· Prime symbol in G’ or R’ is very small and almost invisible in printed form
· Rtask is denoted as a “collection” in Table 1, as “set” in section 3.2
· [3.1 Task formulation] “Task” is overloaded in the paper. “Problem” should be used instead.
· [line 258] Closing brace before vertical line in set comprehension formula should be added.
· [line 384] “TransformerBlcok”
· [Algorithm 1] Title is on page 14, but the algorithm itself is on page 15.
Redundant hyphens should be removed.
Terminological issues like relation/relationship, set/collection, task/problem, etc. should be reconciled.
Author Response
Response to Reviewer 3 Comments
Point 1: Experiments for 6 models from 11 are not reproduced, results are just taken from [34].
Response 1: In response to this issue, we have re-implemented six models using their optimal parameter settings and updated our experimental data accordingly. In Section 5.6, Impact of Few-Shot Size, we included a comparative experiment with the GMatching(MaxP) model and updated Figure 7 while providing a description of the experiments.
Point 2: Implementation details of other 5 models are not mentioned. Do the respective authors provide source code of their models?
Response 2: The implementation details of these five models are described in their respective papers. In the related work section, we have briefly discussed the advantages and limitations of these models. While the authors have provided their model source code, it may not be complete. Therefore, we have independently reconstructed and conducted experiments based on the information presented in the papers.
Point 3: The authors do not provide a way to reproduce their results (probably the experimental framework should be published).
Response 3: Our model's experimental framework code cannot be made publicly available due to a pending patent application.
Point 4: G’ is defined as G with removed “all subgraphs of task relationships” in Table 1 and “by removing all task relationships” in 3.2. So subgraphs or just relations?
Response 4: The background knowledge graph G' refers to the subgraph of G obtained by removing all task-specific relations. In section 3.2, we emphasize that in traditional knowledge graph models, the task specific relations may also appear during the pre-training process, which is not in conflict with this definition.
Point 5: [Table 1] h, r, t denotes both triplets end embeddings
Response 5: h, r, and t represent the initial embedding features of the head entity, relation, and tail entity, respectively.
Point 6: [Table 1] (h, t) in description is not related to R’r
Response 6: We have removed R'r from Table 1.
Point 7: [lines 203-204] “The candidate tail entities for each query triplet are constructed based on entity type constraints[18]” – should be clarified.
Response 7: The candidate tail entities for each query triple are constructed based on entity type constraints, following the method proposed in [18]. This construction method ensures that the query triples will match with candidate tail entities that exhibit semantic similarity.
Point 8: No explicit definition of “i-th higher-order neighbor relationship” is provided. An example should be also provided.
Response 8: The concept of the "i-th higher-order neighbor relationship" can be understood by referring to the illustration in Figure 3.
Point 9: Definition of “softmax” function is not referenced explicitly.
Response 9: The "softmax" function is employed to distinguish unimportant neighbor relationships and achieve the goal of filtering. At the same time, the function's role in the paper is explained as follows: “softmax(.)converts the input into a probability distribution of neighbors.”
Point 10: Correspondence of h and ho is not clear.
Response 10: h and ho have already been explained in the text. h represents the central entity, while ho represents the entity at a specific order or level within the higher-order neighborhood.
Point 11: et, and eh are not explained
Response 11: et represents the tail entity neighbors of an entity, while eh represents the head entity neighbors of an entity.
Point 12: “Selective attention” is not commented
Response 12: In the paper, the role of the selection attention mechanism has already been explained. "Then, based on the similarity score of the higher-order neighborhood encoding, the attention mechanism is employed to assign higher attention scores to entities with higher scores."
Point 13: GLU should be put instead of “GRU”
Response 13: We have made modifications to Figure 3.
Point 14: GLU block is neither explained nor referenced. I can not find Ct-1, Ct, xt variables from the figure in the text
Response 14: t-1 represents the previous time step, while t represents the current time step. C represents the neighborhood encoding of the central entity h, and Xt represents the hidden state. We have made the necessary modifications to Figure 3 as requested.
Point 15: eh and ch are used to produce e on the figure. But formula (6) uses h, but not eh.
Response 15: We have made changes to formula 6.
Point 16: [line 322] Formula should be revised somehow. In this form it is syntactically incorrect.
Response 16: We have made modifications to Equation 8. We have changed the wording to "To enhance clarity, we represent the relationship R in the set of neighbor relationships as a vector, as shown in Equation (8):"
Point 17: [line 324] What is Er?
Response 17: Er has already been described in the text. “Where represents the vectorized representation of the i-th neighbor relationship in the set .
Point 18: [Fugure 5] x variables with different indices are used on the figure, but z variables are used in the text.
Response 18: The meaning of the x variable with different indices in Figure 5 is defined in Equation 14. Figure 5 represents the state of the Transformer Learning Framework at a specific time step.
Point 19: [line 383] “Fusion” is not defined.
Response 19: We have made the following changes regarding the "fusion" aspect:
"Next, the and are connected through a residual connection, serving as the input to the Transformer module. Within the Transformer module, the Multi-Head Attention layer and Add&Norm layers are first utilized for learning. Then, the Feed Forward layer, composed of fully connected layers and ReLU activation functions, along with the subsequent Add&Norm layers, introduce non-linearity to the module. This approach aims to enhance the learning process, ultimately yielding the output of the Transformer learner."
Point 20: [line 425] ||z3i||2 represents the norm of z3i, not z
Response 20: We have made changes. “Where represents the norm of vector ”
Point 21: [Table 2] It seems that #Rel is a number of non-few-shot relations, and not the number of all relations as defined in the caption
Response 21: We have made changes. “#Rel. denotes the number of few-shot relations”
Point 22:I doubt that exact percentage of outperforming should be included in the abstract.
Response 22: We have removed the specific percentages in the abstract. The revised sentence is as follows:
"We conducted comparative experiments on the publicly available NELL-One and Wiki-One da-tasets, comparing FRL-KGC with 6 few-shot knowledge graph completion models and 5 tradi-tional knowledge graph completion models for 5-shot link prediction. The results showed that FRL-KGC outperformed all comparison models in terms of MRR, Hits@10, Hits@5, and Hits@1 metrics."
Point 23:“Relation” and “relationship” is used in an interchangeable way. It seems only “relation” should be used. Even in the title of the paper and in the title of the proposed model different terms are used.
Response 23: We have made the correction and replaced all occurrences of "relationship" with "relation" throughout the entire document.
Point 24: Several existing works apply few-shot relation learning for Knowledge Graph Completion. For instance, in [16][20] the same terms are used even in titles. The authors should provide more specific title for their model to distinguish it from the others.
Response 24: Indeed, the GMatching model [16] and the GANA model [20] have distinct titles, terminology, and descriptions. It is clear that these two models are different and have their unique contributions in the field.
Point 25:[Introduction] Texts in [69-80] lines and in [81-90] lines are very similar. The pieces should be combined, duplications should be removed.
Response 25: To improve the accuracy of link prediction, this paper proposes a few-shot knowledge graph completion model (FRL-KGC), which makes the following contribu-tions:
(1) We introduce the FRL-KGC model, which incorporates a gating mechanism to ex-tract valuable contextual semantics from the head entity, tail entity, and neighborhood in-formation, specifically addressing high order neighborhood information in the knowledge graph. Furthermore, we leverage the correlation between entity pairs in the reference set to represent relations, reducing the dependency of relation embeddings on the central entity.
(2) We effectively utilize both the structural and textual information of the knowledge graph to capture features related to few-shot relations.
(3) Experimental evaluations are conducted on 2 publicly available datasets, and the results demonstrate that our proposed model outperforms other KGC models. Addition-ally, ablation experiments validate the effectiveness of each key module in our model.
Point 26: [Introduction, 60-68] “the high-order neighborhood entity information” and “the influence of entity pairs on the contextual semantics” are not understandable within the Introduction. Additional explanations and references are required.
Response 26: In the FKGC task, the aforementioned models have achieved good results. However, these methods still have some limitations: 1) FAAN fails to effectively utilize the valuable information from high order neighboring entities (the most relevant high order neighbor-hood set in nonlocal graphs[21]) and cannot differentiate the importance of different neighbor information, leading to noise related issues. 2) FSRL simply uses a recurrent au-toencoder to aggregate a small reference set. However, during the training process, he FSRL model tends to excessively depend on entity embeddings, leading to overfitting of relations and a decline in the generalization capability of the model. 3) FSRL does not consider the translational property of the TransE model during matching queries, which can lead to a decline in matching accuracy. 4) Previous models have not adequately con-sidered the impact of entity pairs on contextual semantics[21], resulting in reduced accu-racy in link prediction.
Point 27: [Section 2.1] “Translation-based methods treat relations as translation operations between entity pairs” – should be explained.
Response 27: The following modification has been made: " The approach based on translation methods treats relations as translation operations between entity pairs, where modeling relations is viewed as a form of translation in a low dimensional entity representation space. The existence of associations between entities and relations is determined using a distance scoring function. "
Point 28: [Section 2.3] Methods from this section are not presented in experiments. It should be explained why.
Response 28: In this paper, the authors categorize the Neural Network-based Methods as a class within the traditional knowledge graph completion models. Moreover, the paper includes comparative experiments that involve comparing our method with the DistMult and ComplEx models.
Point 29: [Section 3] Table 1 does not help much if placed in this section. Terminology in description is somehow understandable only after reading the respective sections of the paper.
Response 29: Due to the fact that the meaning of each symbol is only described at its initial appearance, subsequent references to symbols may still occur throughout the paper. Therefore, in order to assist readers in quickly finding the meaning of each symbol, Table 1 has been provided, which describes all the symbols defined in the paper.
Point 30: [lines 199-200] “The associated triplets are randomly divided into a reference set Sr, a query set Qr, and a set Dr={Sr, Qr}”: triplets are divided into Sr and Qr. Dr is just a pair of these sets.
Response 30: Dr represents the set composed of the reference set Sr and the query set Qr.
Point 31: The text of the whole paper should be carefully checked for redundant hyphens like “con-tained” or “en-tities” in abstract.
Response 31: We have proofread the article and removed any unnecessary hyphens.
Point 32: Prime symbol in G’ or R’ is very small and almost invisible in printed form.
Response 32: We have adjusted the display size of prime symbols in the equations to facilitate clearer printing.
Point 33: Rtask is denoted as a “collection” in Table 1, as “set” in section 3.2
Response 33: We have made corrections to Table 1.
Point 34: [3.1 Task formulation] “Task” is overloaded in the paper. “Problem” should be used instead
Response 34: We have replaced the term "Task" with "Problem" throughout the article.
Point 35: [line 258] Closing brace before vertical line in set comprehension formula should be added.
Response 35:
Point 36: [line 384] “TransformerBlcok”
Response 36: The output from the Transformer Block learner serves as the final relationship representation for the task triplet , denoted as .
Point 37: [Algorithm 1] Title is on page 14, but the algorithm itself is on page 15.
Response 37: We have made corrections to this issue.
Point 38: Terminological issues like relation/relationship, set/collection, task/problem, etc. should be reconciled.
Response 38: We have proofread and made changes to the article.

Round 2
Reviewer 1 Report
While the author's responses have clarified the differences in science to the paper, the actual structure and the language of the paper are still quite similar in many places, including the source of the example. While the scientific contribution is acceptable, I'd still recommend that improvements be made to distinguish the papers.
English is improved, however typos still remain. I imagine it should be meta test, not meat test, for example.
Author Response
Point 1: While the author's responses have clarified the differences in science to the paper, the actual structure and the language of the paper are still quite similar in many places, including the source of the example. While the scientific contribution is acceptable, I'd still recommend that improvements be made to distinguish the papers.
Response 1: To highlight the distinctiveness of the paper in terms of its overall structure and language compared to other articles, we have made the following modifications.
In the introduction, we have added an example description of long-tail relations. “For example, in Wikidata, approximately 10% of relations have fewer than 10 triples. Furthermore, in most practical applications such as recommendation systems, social media networks, knowledge graphs undergo dynamic changes over time.”
In the introduction, we have added a transitional sentence to emphasize the novelty of our research. “To overcome the limitations of existing methods, we propose utilizing high order neighborhood entity information to represent each few-shot relation. By considering relations, our FKGC model can infer missing facts more effectively. This approach enhances the model's generalization capability and allows for the utilization of more contextual semantics to handle few-shot relations.”
In "4.2. Relation Representation Encoder," we have made changes to the description to emphasize the distinctive features of this module. “MetaR only represents few-shot relations by averaging the embeddings of all entity pairs in the support set, without considering the correlation between entity pairs in the reference set. Additionally, in previous methods, FSRL simply used a cyclic autoencoder to aggregate a small reference set. However, as training deepens, the model's relation embeddings become overly reliant on entities, leading to relation overfitting and reduced generalization capability.
In this paper, we utilize the neighbor relations of entity pairs in the reference set to enrich the semantic representation of the current relation and reduce the reliance on relation embeddings for entities, thereby enhancing the model's generalization ability.”
In the experimental section, we compared our model with existing baseline models. The existing models can be categorized into two main types: traditional knowledge graph completion models and few-shot knowledge graph completion models. Additionally, we conducted new experiments specifically focusing on traditional knowledge graph completion methods. The experimental conclusion is as follows:
The 5-shot link prediction results of all models on the NELL-One and Wiki-One da-tasets are shown in Table 3. It can be observed from Table 3 that:
1) Compared with traditional knowledge graph embedding methods, FRL-KGC achieves the best performance on both datasets. The experimental results demonstrate that FRL-KGC can effectively predict missing entities in few-shot relations.
2) On both datasets, the FRL-KGC model outperforms the best results of the baseline models on 4 evaluation metrics. Compared with the best-performing MetaR (In-train) model on the NELL-One dataset, the FRL-KGC model improves the MRR, Hits@10, Hits@5, and Hits@1 metrics by 2.9%, 1.9%, 3.1%, and 4.3%, respectively. The performance im-provements on the Wiki-One dataset are 3.3%, 4.3%, 3.4%, and 3.2%, respectively. It is worth noting that only one setting in either Pre-train or In-train can perform well on a sin-gle dataset. This indicates that our model has better generalization ability across different datasets. Furthermore, FRL-KGC can leverage the contextual semantics and structural in-formation of entities in KGs to improve the performance of few-shot knowledge graph completion.
In the conclusion section, we have made the following changes to emphasize our contributions and provide insights into future research directions. “The experimental results indicate that the FRL-KGC model outperforms existing FKGC models in terms of link prediction accuracy. However, the design of the dataset does not fully capture the dynamic nature of few-shot knowledge graphs. If the knowledge graph undergoes real-time changes, ensuring the model's inference accuracy becomes a chal-lenging problem. In our future work, we plan to explore the use of timestamps to enhance the model's representation capacity of the knowledge graph and maintain inference ac-curacy in dynamic few-shot knowledge graph learning. Additionally, we will investigate the use of external knowledge sources to augment the representations of entities and rela-tions, such as leveraging textual descriptions of entities and relations.”
Point 2: English is improved, however typos still remain. I imagine it should be meta test, not meat test, for example.
Response 2: We have checked the spelling and corrected the errors. "meat test" has been changed to "meta test."
Point 3: Please check that all references are relevant to the contents of the manuscript.
Response 3: We have rechecked all the references cited in the manuscript. The references cited in the introduction, related work, and comparison model sections have been thoroughly reviewed. All the references cited in the manuscript are relevant to the research content.
